# A Six-Year Prospective Study on Problem Drinking among Evacuees of the Great East Japan Earthquake: The Fukushima Health Management Survey

**DOI:** 10.3390/ijerph20010319

**Published:** 2022-12-25

**Authors:** Yuka Ueda, Fumikazu Hayashi, Tetsuya Ohira, Masaharu Maeda, Seiji Yasumura, Itaru Miura, Shuntaro Itagaki, Michio Shimabukuro, Hironori Nakano, Kenji Kamiya, Hirooki Yabe

**Affiliations:** 1Department of Neuropsychiatry, Fukushima Medical University School of Medicine, Fukushima 960-1295, Japan; 2Radiation Medical Science Center for the Fukushima Health Management Survey, Fukushima Medical University, Fukushima 960-1295, Japan; 3Department of Epidemiology, Fukushima Medical University School of Medicine, Fukushima 960-1295, Japan; 4Department of Disaster Psychiatry, Fukushima Medical University School of Medicine, Fukushima 960-1295, Japan; 5Department of Public Health, Fukushima Medical University School of Medicine, Fukushima 960-1295, Japan; 6Department of Diabetes, Endocrinology and Metabolism, Fukushima Medical University School of Medicine, Fukushima 960-1295, Japan

**Keywords:** problematic drinking, disaster, evacuees, epidemiology, risk/protective factors

## Abstract

Evacuees of the Great East Japan Earthquake have experienced adverse, long-term physical and psychological effects, including problem drinking. This study examined the risk and recovery factors for problem drinking among evacuees between fiscal years (FY) 2012 and 2017 using data on residents in the evacuation area from the Mental Health and Lifestyle Survey. With the FY 2012 survey as a baseline, a survey comprising 15,976 men and women was conducted in the evacuation area from FY 2013 to FY 2017, examining the risk and protective factors for problem drinking. Particularly, the Cutting down, Annoyed by criticism, Guilty feeling, and Eye-opener (CAGE) questionnaire was used to evaluate problem drinking. Univariate and multivariate Cox proportional hazard models were constructed to identify the risk and recovery factors of problem drinking. The findings indicated that the male gender, insufficient sleep, job change, trauma symptoms, mental illness, family financial issues, and heavy drinking (≥4 drinks per day) were significant risk factors for the incidence of problem drinking among the evacuees. Furthermore, a high blood pressure diagnosis could exacerbate problem drinking among men, while younger age and a diabetes mellitus diagnosis could increase problem drinking among women. Trauma symptoms and heavy drinking inhibited recovery from problem drinking after the disaster. Understanding these factors can shape effective long-term intervention strategies to physically and psychologically support evacuees.

## 1. Introduction

Research has shown that post-traumatic stress after natural disasters is linked to an elevated risk of problem drinking [1] and substance abuse such as cigarette use [2,3]. Numerous studies have reported that man-made and natural disasters, as well as terrorist attacks, are associated with increased alcohol consumption [3,4,5]. Alcohol dependence after a disaster is related to poor mental health among affected individuals [6]. Particularly, research has indicated increased alcohol consumption among several evacuees who use it as “self-medication” to mitigate their symptoms after experiencing trauma [7,8]. Furthermore, severe symptoms of post-traumatic stress disorder (PTSD) have been strongly associated with alcohol use for “coping-motivated drinking” following exposure to a disaster [9]. The Great East Japan Earthquake and Tsunami, which occurred on 11 March 2011, is considered a compound disaster, as it resulted in the exacerbated abuse of psychoactive substances, particularly alcohol, high smoking prevalence [10], and impaired sleep quality [11]. A previous study showed that evacuees changed their drinking behavior after experiencing a compound disaster. Moreover, the study also found that individuals who began drinking after the disaster had a higher risk of developing severe mental illness [12]. Another study contrasted the differences and similarities among risk factors in the development of problem drinking based on gender [13]. Furthermore, research has indicated that sleep insufficiency and heavy drinking culminated in problem drinking in both genders. Particularly, family finances and severe trauma symptoms caused problem drinking among male evacuees, while a history of mental illness increased problem drinking among female evacuees.

Based on these research findings, the present study examined the risk and protective factors for problem drinking within the context of the Great East Japan Earthquake and Tsunami of 11 March 2011 [12,13]. This study assessed how evacuees developed or recovered from problem drinking based on the Cutting down, Annoyed by criticism, Guilty feeling, and Eye-opener (CAGE) score for six years. A previous study [5] gathered over 15 years of data after the 9/11 terror attacks on the World Trade Center (WTC) in New York City and found that binge drinking was strongly linked with the PTSD symptom cluster. Furthermore, the study noted that alcohol was used intentionally as a means of self-medication. Another study found that traumatic stress may be associated with problem drinking after prolonged exposure, and that men and younger people were more likely to begin problem drinking two years after 9/11 [14]. Most studies to date have looked at short-term impacts in the first 2 years following the disaster, with few looking at the longer term. We considered the importance of conducting a long-term longitudinal study for the evacuees of the Great East Japan Earthquake in 2011. Our study hypothesized that psychological distress, trauma symptoms, and insufficient sleep culminated in developing problem drinking between the fiscal years (FY) 2012 and 2017, using a longitudinal design as a measure. It further hypothesized that social networks or support facilitate recovery from problem drinking. This is the first prospective study to explore and present the risks and protective factors for problem drinking among evacuees of the Great East Japan Earthquake.

## 2. Materials and Methods

### 2.1. Data Source

This study used data from the Mental Health and Lifestyle Survey that assessed the mental health and lifestyle of residents of evacuation areas after the Great East Japan Earthquake. The complete survey protocol was published in FY 2012 [15]. The target participants for the survey lived in 13 municipalities: Hirono, Naraha, Tomioka, Kawauchi, Okuma, Futaba, Namie, Katsurao, Iitate, Tamura, Minami-Soma, Kawamata, and Date, which consist of designated areas for evacuation allocated by the government at the time of the accident. Only the municipalities of Minami-Soma, Tamura, and Kawamata included evacuees and non-evacuees. These residents received questionnaires annually from 18 January 2012 [15,16]. Data from FY 2013 to FY 2017 were used to elucidate the development of problem drinking for six years after the disaster. The participants of the mental health survey were informed that the survey results would be examined and reported after analysis. Furthermore, only the individuals who returned the self-recorded questionnaire were considered to have provided consent to participate in the survey. This study was approved by the ethical review board of Fukushima Medical University (approval number: 2020-239).

### 2.2. Study Sample

Figure 1 presents the participant flow chart. The target population comprised 43,990 adults, aged 20 years and above, who responded to the FY 2012 survey (response rate 19.9%, *n* = 184,507). The sample population excluded those who did not answer the CAGE questionnaire in FY 2012 (*n* = 9899) and who did not respond to the questionnaire independently (*n* = 2547), because CAGE can only be assessed through self-response. Furthermore, respondents with a CAGE score of ≥2 (*n* = 3241) in FY 2012 were excluded. The remaining 27,064 respondents (men: 12,120; women: 14,944) formed the baseline sample for follow-up. Likewise, those individuals who did not respond on their own to the assessment from FY 2013 to FY 2017 (*n* = 620), and did not respond to the follow-up (*n* = 10,468), were excluded. Thereafter, the longitudinal data were obtained for 15,976 participants (men: 9117; women: 6859).

Figure 2 presents a flow chart displaying the protective factors that helped in recovery from problem drinking. This flow chart excluded participants whose CAGE scores were below 2 (*n* = 27,064) in FY 2012. The remaining 3241 respondents (men: 2482, women: 759) formed the baseline sample for follow-up. Similarly, individuals who did not respond to the questionnaire independently from FY 2013 and FY 2017 (*n* = 61), or who did not respond to the follow-up (*n* = 373) were excluded from the sample. Thereafter, longitudinal data were obtained from 2807 participants (men: 2224, women: 583).

### 2.3. Measures

This study evaluated all variables used previously [12,13], including alcohol consumption and problem drinking (CAGE), general and socio-demographic status variables, current social network status, sleep insufficiency, risk of serious mental illness and psychological distress (K6), and trauma symptoms (PCL-S).

#### 2.3.1. Alcohol Use and Problem Drinking Measures

To align with previous studies [12,13], heavy drinking/alcohol consumption that would enhance the risk of a lifestyle disease was defined as having four or more drinks per day (≥44 g of ethanol). According to this definition, a drink could comprise 120 mL of spirit (e.g., whiskey or brandy), 480 mL of wine, 1000 mL of beer, or 360 mL of sake. This definition is consistent with the reported median for moderate and proper drinking (20 g of ethanol per day) and heavy drinking (60 g of ethanol per day) [17].

The CAGE questionnaire is used to screen for alcohol dependency [18] and diagnose alcoholism [19]. The validity and reliability of the screening test have been confirmed. Furthermore, providing at least two positive answers was classified as indicating alcohol dependence, irrespective of the respondent’s sex [18]. Therefore, a CAGE score of ≥2 represented a drinking problem.

#### 2.3.2. Socio-demographic Variables

This study evaluated various demographic characteristics, socio-economic factors, and disaster-related risk factors related to problem drinking [14,20,21]. Demographic factors were obtained, including sex, age (i.e., 20–49, 50–64, or ≥65 years), and history of a diagnosed of any mental illness, hypertension, and diabetes mellitus (i.e., Yes or No) [13]. This study also assessed socio-demographic factors such as employment change (i.e., change in work before and after the disaster) and post-disaster family financial situation (i.e., severe, below average, average, not severe) [13].

#### 2.3.3. Current Social Network Status

The Lubben Social Network Scale (LSN-6) was used to screen current social networks, including family and friends, among the evacuees [22,23]. The validity of this test was explained in a previous study that used the Japanese version of the LSNS-6 [13].

#### 2.3.4. Sleep Insufficiency

Research has shown that individuals face sleep problems after traumatic events [24]. An earlier study has shown that sleep insufficiency is a risk factor for problem drinking; thus, the participants were asked the same questions [13].

#### 2.3.5. Risk of Serious Mental Illness and Psychological Distress

This study used the six-item Kessler Psychological Distress Scale (K6) to screen for non-specific serious mental illnesses [25], in which scores ranging from 13 to 24 are classified as “probable serious mental illness” [26]. The validity of the Japanese version of the K6 has been explored in previous studies [12,13,27].

#### 2.3.6. Trauma Symptoms

Comorbidity of PTSD and problem drinking has been well-documented [4,28]. Therefore, this study used the PTSD Checklist-Specific (PCL-S) to identify traumatic symptoms among evacuees. Consistent with previous research [29], a cut-off of 44 was used to diagnose PTSD. The validity of the Japanese version of the PCL-S [30,31] was explored in a previous study [13].

### 2.4. Data Analysis

This study analyzed data for six years, from FY 2012 to FY 2017, using univariate Cox proportional models to investigate possible predictors of problem drinking. The analysis used FY 2012 as the baseline and FY 2013 to FY 2017 as the follow-up period. For participants who had multiple visits during the follow-up, the most recent data were used in the analysis as the follow-up results. The dependent variables were chosen based on previous studies [2,20] and were employed as multivariate adjustment variables. Missing data were complemented with dummy variables.

All statistical analyses were performed using SAS 9.4 (SAS Institute Inc., Cary, NC, USA). Univariate and multivariate Cox proportional hazards models were used to obtain crude and adjusted hazards ratios (HRs) and 95% confidence intervals (Cis) for the association between each factor and problem drinking or recovery. Multivariate Cox proportional hazards models for men and women were established to determine differences based on gender. *p* < 0.05 indicated statistical significance.

## 3. Results

Table 1 shows the breakdown of variables according to change (or a lack thereof) in problem drinking (i.e., individuals with low to high scores were emerging problem drinkers; individuals with continuous low scores were current non-problem drinkers) from FY 2012 to FY 2017. The table also highlights motivating factors of characteristics associated with these changes. In total, there were 1949 emerging problem drinkers and 11,463 current non-problem drinkers. Furthermore, 14.5% of the participants developed problem drinking between FY 2012 and FY 2017. Moreover, a higher number of men than women developed problem drinking during the study period. Emerging problem drinkers included a higher proportion of those with K6 and PCL-S scores of ≥13 and ≥44, respectively, and they consumed more alcohol than the current non-problem drinkers. Becoming a problem drinker was associated with age, subjective health condition, history of a serious mental illness, sleep insufficiency, high blood pressure, diabetes mellitus, and family financial status (*p* < 0.05).

Table 2 presents an overview of the univariate Cox proportional hazard models, which were established using factors deemed significant during survival analysis to identify the association between social and psychological indicators and problem drinking frequency among the evacuees. Table 2 shows that being a man (HR: 2.30; 95% CI: 2.09–2.53), heavy alcohol consumption (HR: 2.02; 95% CI: 1.85–2.21), sleep insufficiency (HR: 1.64; 95% CI: 1.46–1.84), psychological distress (HR: 1.63; 95% CI: 1.43–1.85), trauma symptoms (HR: 1.98; 95% CI: 1.79–2.19), a history of mental illness (HR: 1.52; 95% CI: 1.32–1.76), and family finances (HR: 1.74; 95% CI: 1.57–1.94) significantly influenced the development of problem drinking.

Table 3 presents the results of multivariate Cox proportional hazards analysis according to sex to determine sex-based differences. Alcohol consumption, trauma symptoms, and family finances were common risk factors for problem drinking among both men and women. Sleep insufficiency (HR: 1.22; 95% CI: 1.04–1.42) and high blood pressure (HR: 1.12; 95% CI: 1.00–1.25) significantly influenced problem drinking among men, independent of age. In contrast, younger age (HR: 1.59; 95% CI: 1.21–2.08), a history of diabetes mellitus (HR: 1.32; 95% CI: 1.02–1.69), and a history of mental illness (HR: 1.31; 95% CI: 1.02–1.69) were significant risk factors for problem drinking among women independent of age.

This study also examined the factors that enabled recovery from problem drinking between FY 2012 and 2017. Table 4 presents the variables based on the change (or a lack thereof) in recovery rates from problem drinking (i.e., individuals with high to low scores were recovering problem drinkers; individuals with continuously high scores were current problem drinkers) from FY 2012 to 2017, revealing the characteristics associated with these changes. The total number of recovering and current problem drinkers was 1993 and 814, respectively. Furthermore, 71.0% of the participants were recovering problem drinkers from FY 2013 to FY 2017. Recovering problem drinkers also included a higher proportion of those with K6 and PCL-S scores of <13 and <44, respectively, and were not heavy drinkers when compared to current problem drinkers. Recovery from problem drinking was associated with age, subjective health, sleep insufficiency, family financial status, and alcohol consumption (*p* < 0.05).

Table 5 presents an overview of univariate Cox proportional hazards models, which were established using factors that were determined as significant during survival analysis, to identify the association between social and psychological indicators and frequency of recovery from problem drinking among the evacuees. Univariate Cox proportional hazards analysis showed that LSN-6 tended to be associated with a reduced risk of problem drinking (HR: 0.93; 95% CI: 0.85–1.02) although its statistical significance was not found.

Table 6 presents the results of multivariate Cox proportional hazards analysis according to sex to determine sex-based differences. Heavy drinking (≥4 drinks) and trauma symptoms (PCL ≥ 44) were significant factors that prevented recovery from problem drinking.

## 4. Discussion

This study examined the risk and recovery factors for problem drinking from FY 2012 to FY 2017 among evacuees from regions affected by the Great East Japan Earthquake. The results showed that there are some similarities and differences between men and women in developing problem drinking after disasters. Additionally, heavy drinking (≥4 drinks) and trauma symptoms (PCL ≥ 44) were found to be significant factors that prevented recovery from problem drinking among both genders. Previous studies have reported that individuals are impacted by drinking behavior, including alcohol consumption for two years after traumatic events [6,32,33,34]. However, our study is the first to underscore the risk factors and recovery from problem drinking over six years.

This research found that a substantial proportion of the sample (15.5%) developed problem drinking within six years of experiencing a compound disaster. This implies that several evacuees still suffered from disaster-related drinking problems for more than a few years after the disaster. Therefore, one must provide seamless support for evacuees who suffer from drinking issues by understanding any risk factors such as trauma issues and any other risk factors underlying problem drinking. The results also show that alcohol consumption (≥4 drinks), disaster-related factors, family finances, and sleep insufficiencies were related to the development of problem drinking from FY 2012 to FY 2017, which were the same risk factors in the chronic post-disaster phase from FY 2012 to FY 2013. Additionally, heavy drinking was found to be a significant factor in the development of problem drinking from FY 2012 to FY 2017 among men and women. Risk factors for the development of problem drinking in the chronic phase after a compound disaster, such as male sex, sleep insufficiency, trauma symptoms (PCL-S ≥ 44), and family finances, were constant from FY 2012 to FY 2013. Notably, upon comparing the HR in the short-term research from FY 2012 to FY 2013, the HR of men with problem drinking in the present study from FY 2012 to FY 2017 (2.03 95% CI: 1.83–2.04) was higher than the odds ratio (OR = 1.77, 95%CI: 1.41–2.21) in the previous study [12]. Furthermore, the HRs for trauma symptoms (PCL-S ≥ 44) and alcohol consumption (≥4 drinks) remained high between the chronic phase and FY 2012–FY 2017. Therefore, both alcohol consumption and trauma symptoms led to the development of problem drinking. Notably, continued trauma symptoms and heavy alcohol consumption could comprise severe risk factors for developing problem drinking among men in the period from FY 2012 to FY 2017 in contrast to FY 2012 to FY 2013. Thus, the findings of this study emphasize the importance of (a) having practitioners intervene to support evacuees who have a drinking problem, (b) paying attention to the assessment of trauma symptoms, and (c) providing psychoeducation on alcohol. Meanwhile, a mental illness diagnosis and subjective health conditions indicated long-term risk factors for problem drinking, but not in the short term.

Variation based on gender was found in the risk factors for developing problem drinking from FY 2012 to FY 2017. Sleep insufficiency and high blood pressure were significant risk factors in men. Furthermore, women of a younger age (i.e., 20–49 years) and diagnosed with diabetes mellitus and mental illness constituted significant risk factors. An association between physical illnesses such as high blood pressure and diabetes mellitus and the development of problem drinking among evacuees is a new finding. Moreover, insomnia has been strongly associated with problem drinking [35,36] as alcohol has also been used as a medication for insomnia [35].

Assessing the differences in risk factors based on gender is crucial for understanding how problem drinking develops and when particular intervention plans can be implemented. In contrast to the preceding short-term study between FY 2012 and FY 2013, this study found differences based on gender. A history of diabetes mellitus and mental illness was significantly associated with the risk of problem drinking among women. To examine whether the result is a reversal of causality, the results of the follow-up study, excluding the FY 2013 data, were analyzed, as shown in the Appendix A (Table A1, Table A2, Table A3, Table A4, Table A5 and Table A6). The results showed that diabetes mellitus was still a risk factor among women (HR: 1.47; 95% C1: 1.11–1.94). Therefore, this study statistically analyzed the breakdown of women both with and without a history of diabetes mellitus for each response to CAGE questions. Positive responses to three questions (i.e., “Have you ever felt you ought to cut down on your drinking?” Have people annoyed you by criticizing your drinking?”, and “Have you ever felt bad or guilty about your drinking?”) were high among women with a history of diabetes mellitus (*p <* 0.05). The results may indicate that women who were diagnosed with diabetes mellitus with problem drinking had a high tendency of having felt that they ought to cut down on their drinking and felt guilty of their drinking behavior. A study previously reported that evacuees experienced difficulty in accessing medication, treatment, and clinical services [37]. Particularly, the women evacuees diagnosed with diabetes mellitus might have encountered difficulty in seeking treatment for their drinking problems as they felt guilty for their behavior.

Thus, this study analyzed factors that prevented the development of problem drinking from FY 2012 to FY 2017 for both genders. Unfortunately, the current results did not reveal any specific protective factor. However, it was found that trauma symptoms and alcohol consumption prevented recovery.

Therefore, the LSN-6 may be a key protective factor for evacuees with problem drinking. Research has also presented disaster research on the Great East Japan Earthquake, suggesting that continuous intervention for evacuees with alcoholism who lived alone in temporary housing helped them recover from their drinking problems [38]. Thus, instead of isolating evacuees suffering from drinking problems, it is important to provide continuous support.

Based on current knowledge, this is the first study to examine risk and protective factors for problem drinking among evacuees affected by the Great East Japan Earthquake between FY 2012 and FY 2017. Particularly, this study compared the risk factors for problem drinking between the period from FY 2012 to FY 2013 and FY 2012 to FY 2017 among men and women. Consequently, problem drinking was found to be caused by physical, psychological, and economic crises, and risk factors increased substantially between FY 2012 and FY 2017. This suggests that medical practitioners should implement long-term interventions to support evacuees with problem drinking habits.

Meta-analyses and population-based studies have demonstrated how alcohol consumption changes after a traumatic event and/or with risk factors [8,39]. However, some studies have examined both problem drinking and alcohol consumption vis-à-vis risk factors after compound disasters in the long term. Therefore, this is the first study to examine how alcohol consumption and the main risk factors, such as socio-demographic variables, sleep insufficiency, psychological distress (K6), trauma symptoms (PCSL-S), and alcohol consumption, culminate in problem drinking.

This study has several limitations. First, the response rate was 19.9% in FY 2012. Therefore, the results may have overestimated or underestimated the impact of problem drinking after the Great East Japan Earthquake. Second, a previous study explained that the persistence of feelings such as helplessness and hopelessness due to grief among evacuees was a predictor of increased drinking [32]. However, as these factors were not evaluated in the present study, there may be confounding unadjusted latent factors that contribute to the risk of problem drinking. Finally, this study used Cox analysis to identify risk and protective factors from FY 2012 to FY 2017. Thus, the individual changes in the CAGE scores were not analyzed. Thirdly, problem drinking was assessed in this study using a standard questionnaire, CAGE, while sleep was assessed solely based on participants’ subjective symptoms

## 5. Conclusions

This study found that recovery from heavy alcohol consumption and alleviation of trauma symptoms are key factors in enabling recovery from problem drinking among evacuees. It contributes to the literature by identifying risk and protective factors for problem drinking in the long term. Understanding these factors can shape effective long-term intervention strategies to physically and psychologically support evacuees. The practitioners are required to continue to have long-term, large-scale surveys, to conduct follow-up interventions for the evacuees who have suffered from a drinking problem.

Therefore, the government needs funding to provide long-term support for evacuees recovering from alcohol addiction.

## Figures and Tables

**Figure 1 ijerph-20-00319-f001:**
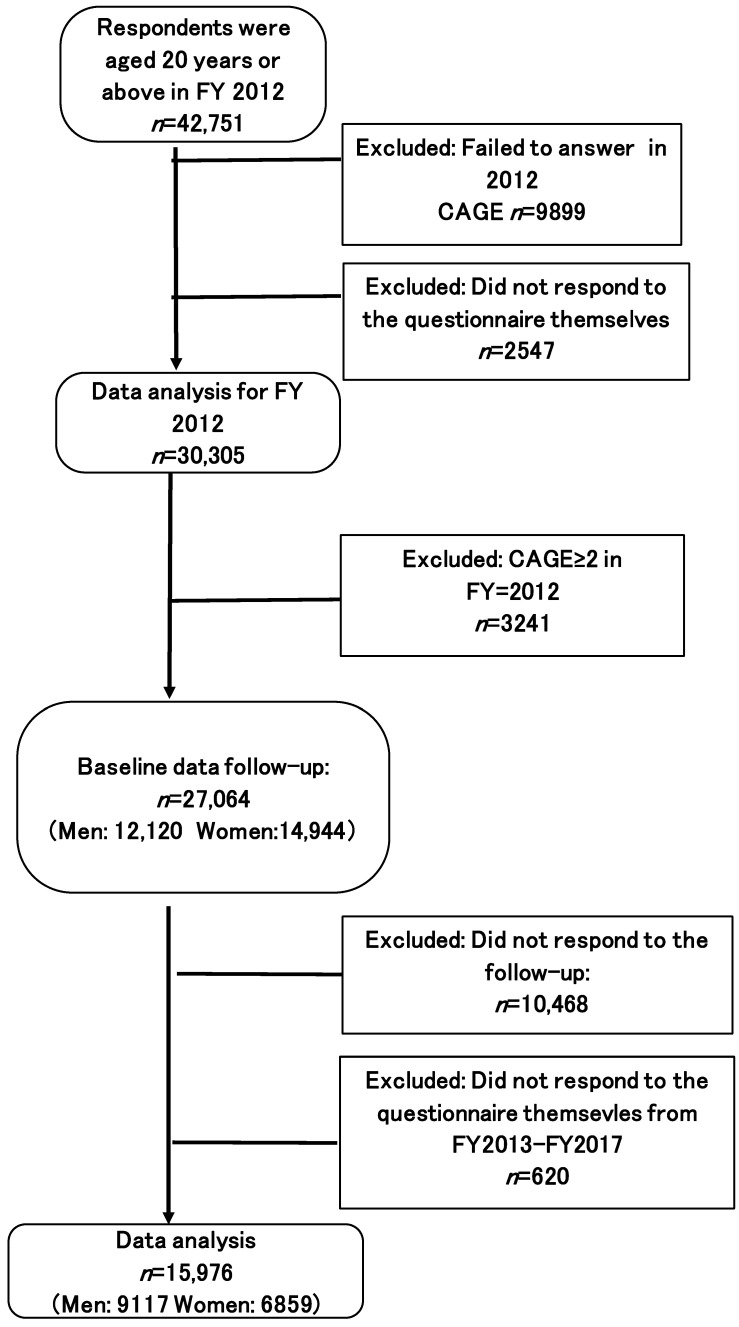
Participant flow chart (CAGE < 2 in 2012).

**Figure 2 ijerph-20-00319-f002:**
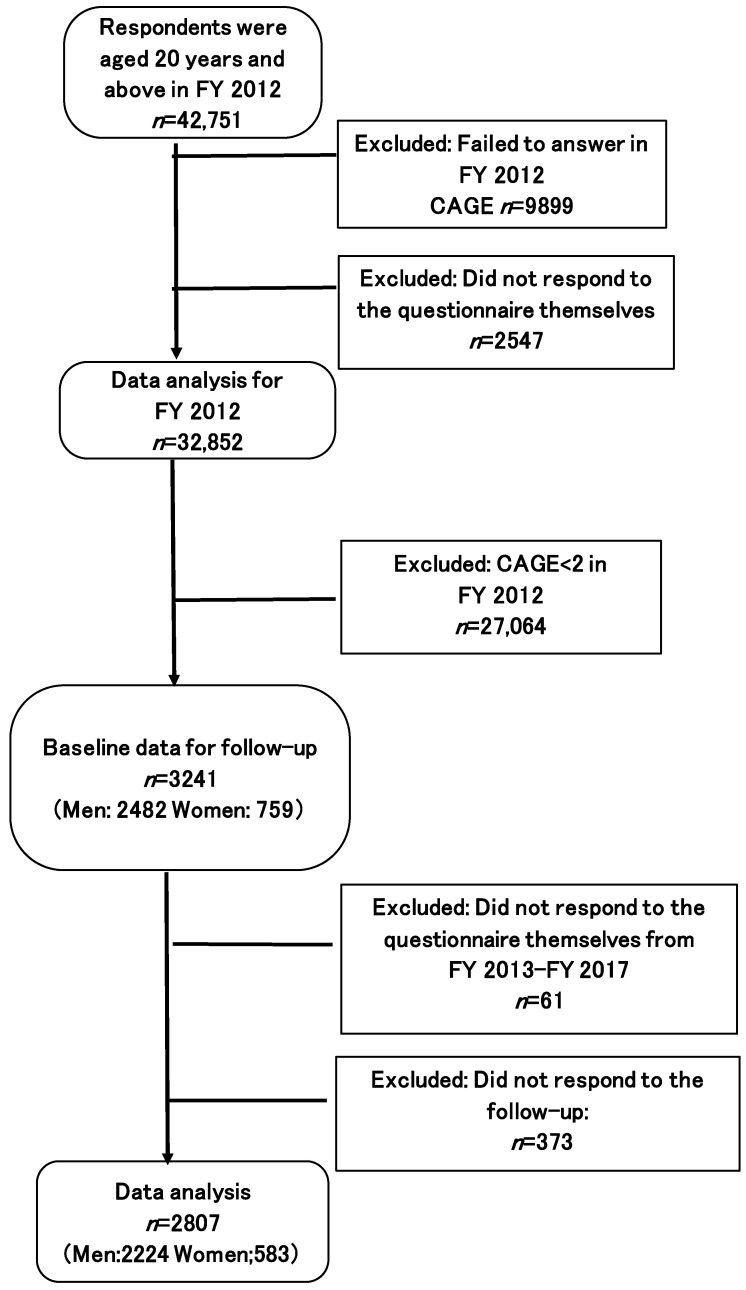
Participant flow chart (CAGE ≥ 2 in 2012).

**Table 1 ijerph-20-00319-t001:** Characteristics of participants who only developed problem drinking after FY 2012.

	Total	Maintaining Non-Problem Drinkers CAGE < 2 in 2012→CAGE < 2 in 2017	%	Emerging Problem Drinkers CAGE < 2 in 2012→CAGE ≥ 2 in 2017	%	*p*
Sex						
Men	9117	7251	53.7	1866	75.6	<0.0001
Women	6859	6256	46.3	603	23.4	
Age						
20–49 years	5554	4580	33.9	974	39.4	
50–64 years	5602	4729	35	873	35.4	<0.0001
≥65 years	4820	4198	31.1	622	25.2	
Subjective health condition						
Very good–Good	3527	3080	23.4	447	18.6	<0.001
Normal	9801	8281	62.8	1520	63.4	
Poor–Very poor	2256	1825	13.8	431	18.0	
History of diagnosed mental illness						
No	14,676	12,466	93.8	2210	91.5	<0.001
Yes	1025	820	6.2	205	8.5	
Diagnosed with high blood pressure						
No	8815	7671	57.3	1144	47	
Yes	7020	5728	42.7	1292	53.0	<0.001
Diagnosed with diabetes mellitus						
No	12,492	10,699	80.5	1793	74.6	
Yes	3201	2589	19.5	612	25.4	<0.001
Diagnosed with hyperlipidemia						
No	9531	8119	61.1	1412	58.4	
Yes	6169	5165	38.9	1004	41.6	0.013
Exercise frequency						
Every day	2252	1858	13.9	394	16.3	0.018
2–4 times a week	3519	2974	22.3	545	22.5	
Once a week	2551	2178	16.3	373	15.4	
None	7434	6325	47.4	1109	45.8	
Sleep insufficiency						
Satisfied	6004	5163	39.3	841	35.9	<0.001
A little dissatisfied	7102	6007	45.7	1095	45.6	
Very dissatisfied to quite problematic	2440	1974	15	466	19.4	
Employment change						
Yes	998	822	6.1	176	7.1	0.049
No	14,978	12,685	93.9	2293	92.9	
Family finances						
Severe	2395	1875	14.7	520	22.4	<0.001
Below average	4623	3870	30.2	753	32.4	
Average	7519	6529	51	990	42.6	
Not severe	583	521	4.1	62	2.7	
Psychological distress						
K6 < 13	14,087	11,992	91.9	266	11.3	<0.001
K6 ≥ 13	1328	1062	8.1	2095	88.7	
Trauma symptoms						
PCL < 44	13,072	11,226	87.3	1846	78.7	<0.001
PCL ≥ 44	2127	1626	12.7	501	21.3	
Social network						
LSN_6 < 12	5865	4892	37.5	973	40.8	<0.001
LSN_6 ≥ 12	9459	8139	62.5	1410	59.2	
Alcohol consumption (drinks)						
<4	12,965	11,333	87.3	1632	69.4	<0.001
≥4	2366	1646	12.7	720	30.6	

CAGE: Cutting down, Annoyed by criticism, Guilt, and Eye-opener questionnaire; LSN: Lubben Social Network Scale; PCL: PTSD Checklist-specific; K6: Kessler psychological distress scale.

**Table 2 ijerph-20-00319-t002:** Crude hazard ratios and 95% confidence intervals for the occurrence of problem drinking from FY 2012 to FY 2017.

	All		Men		Women	
	HR	95%CI	HR	95%CI	HR	95%CI
Sex (Reference: Women)						
Men	2.3	2.09–2.53				
Age (Reference: ≥65 years)						
20–49 years	0.99	0.89–1.09	0.98	0.89–1.08	1.44	1.17–1.78
50–64 years	0.98	0.90–1.08	0.84	0.74–0.96	1.09	0.87–1.36
Subjective health condition (Reference: Very good–Good)						
Normal	1.32	1.19–1.47	1.27	1.13–1.43	1.5	1.19–1.90
Poor–Very poor	1.71	1.50–1.96	1.65	1.41–1.91	2.01	1.52–2.67
Exercise frequency (Reference: Every day)						
2–4 times a week	0.96	0.84–1.09	0.99	0.86–1.14	0.86	0.63–1.19
Once a week	0.94	0.81–1.08	0.95	0.81–1.12	0.9	0.64–1.25
None	1.02	0.91–1.16	1.02	0.89–1.17	1	0.74–1.33
History of diagnosed mental illness (Reference: None)						
Yes	1.52	1.32–1.76	1.49	1.25–1.77	1.66	1.28–2.14
Diagnosed with high blood pressure (Reference: None)						
Yes	1.25	1.14–1.37	1.26	1.13–1.39	1.25	1.03–1.53
Diagnosed with diabetes mellitus (Reference: None)						
Yes	1.2	1.09–1.32	1.15	1.04–1.28	1.45	1.15–1.82
Diagnosed with hyperlipidemia (Reference: None)						
Yes	1.02	0.93–1.10	1.06	0.97–1.16	0.94	0.78–1.14
Sleep insufficiency (Reference: Satisfied)						
A little dissatisfied	1.23	1.13–1.35	1.27	1.14–1.40	1.17	0.96–1.42
Very dissatisfied to quite problematic	1.64	1.46–1.84	1.66	1.45–1.90	1.67	1.33–2.09
Employment change (Reference: None)						
Yes	1.23	1.05–1.44	1.2	0.99–1.44	1.33	1–1.77
Family finances (Reference: Average)						
Severe	1.74	1.57–1.94	1.76	1.55–1.98	1.72	1.38–2.15
Below average	1.23	1.12–1.36	1.19	1.06–1.33	1.38	1.14–1.66
Not severe	0.77	0.59–0.99	0.65	0.47–0.89	1.17	0.76–1.79
Psychological distress (Reference: K6 < 13)						
K6 ≥ 13	1.63	1.43–1.85	1.64	1.41–1.92	1.65	1.31–2.07
Trauma symptom (Reference: PCL < 44)						
PCL ≥ 44	1.98	1.79–2.19	2	1.78–2.25	1.98	1.64–2.40
Social network (Reference: LSN_6 ≥ 12)						
LSN_6 < 12	1.17	1.07–1.27	1.15	1.04–1.26	1.24	1.05–1.47
Alcohol consumption (Reference: <4 drinks)						
≥4 drinks	2.02	1.85–2.21	1.88	1.71–2.08	3.25	2.57–4.12

FY: Fiscal year; HR: Hazard ratio; CI: Confidence interval.

**Table 3 ijerph-20-00319-t003:** Multivariate-adjusted HRs and 95% CIs for the occurrence of problem drinking from FY 2012 to FY 2017.

	All		Men		Women	
	HR	95%CI	HR	95%CI	HR	95%CI
Sex (Reference: Women)						
Men	2.03	1.83–2.24				
Age (Reference: ≥65 years)						
20–49 years	1.09	0.96–1.24	0.93	0.78–1.08	1.59	1.21–2.08
50–64 years	0.98	0.88–1.08	0.97	0.87–1.09	1.1	0.86–1.40
Subjective health condition (Reference: Very good–Good)						
Normal	1.15	1.03–1.28	1.11	0.98–1.25	1.3	1.02–1.66
Poor–Very poor	1.14	0.98–1.33	1.09	0.92–1.30	1.28	0.93–1.77
History of diagnosed mental illness (Reference: None)						
Yes	1.16	1.00–1.35	1.12	0.93–1.34	1.31	0.99–1.73
Diagnosed with high blood pressure (Reference: None)						
Yes	1.1	1.00–1.21	1.12	1.00–1.25	1.03	0.83–1.28
Diagnosed with diabetes mellitus (Reference: None)						
Yes	1.07	0.97–1.19	1.03	0.92–1.15	1.31	1.02–1.69
Sleep insufficiency (Reference: Satisfied)						
A little dissatisfied	1.12	1.02–1.24	1.16	1.05–1.29	1.06	0.87–1.29
Very dissatisfied to quite problematic	1.2	1.05–1.37	1.22	1.04–1.42	1.22	0.95–1.58
Employment change (Reference: No)						
Yes	1.17	1.00–1.37	1.15	0.95–1.39	1.27	0.95–1.69
Family finances (Reference: Average)						
Severe	1.36	1.21–1.53	1.39	1.22–1.59	1.3	1.02–1.65
Below average	1.1	1.00–1.22	1.07	0.96–1.20	1.2	0.99–1.46
Not severe	0.82	0.63–1.06	0.68	0.49–0.94	1.26	0.82–1.94
Psychological distress (Reference: K6 < 13)						
K6 ≥ 13	0.94	0.81–1.09	0.94	0.78–1.13	0.94	0.70–1.25
Trauma (Reference: PCL < 44)						
PCL ≥ 44	1.62	1.44–1.83	1.65	1.43–1.90	1.54	1.22–1.95
Social network (Reference: LSN_6 ≥ 12)						
LSN_6 < 12	1.05	0.96–1.15	1.05	0.95–1.16	1.07	0.90–1.27
Alcohol consumption (Reference: <4 drinks)						
≥4 drinks	1.99	1.82–2.18	1.86	1.69–2.05	3.23	2.54–4.10

Multivariate analysis adjusted for sex and age.

**Table 4 ijerph-20-00319-t004:** Characteristics of problem drinkers in FY 2012 by subsequent recovery.

	Total	Current Problem Drinkers	%	Recovery Problem Drinkers	%	*p*
		CAGE ≥ 2 in 2012→		CAGE ≥ 2 in 2012→		
		CAGE ≥ 2 in 2017		CAGE < 2 in 2017		
Sex	2807					
Men	2224	636	78.1	1588	79.7	0.3595
Women	583	178	21.9	405	20.3	
Age						
20–49 years	980	228	28	752	37.7	<0.0001
50–64 years	1038	317	38.9	721	36.2	
≥65 years	789	269	33	520	26.1	
Subjective health condition						
Very good–Good	439	95	12.1	344	17.6	<.0001
Normal	1656	452	57.8	1204	61.5	
Poor–Very poor	645	235	30.1	410	20.9	
History of diagnosed mental illness						
No	2444	679	87.3	1765	90.8	0.006
Yes	277	99	12.7	178	9.2	
Diagnosed with high blood pressure						
No	1265	381	47.7	884	44.8	0.163
Yes	1505	417	52.3	1088	55.2	
Diagnosed with diabetes mellitus						
No	1996	584	73.8	1412	72.5	0.485
Yes	742	207	26.2	535	27.5	
Diagnosed with hyperlipidemia						
No	1589	470	59.9	1119	57.4	0.2244
Yes	1144	314	40.1	830	42.6	
Exercise frequency						
Every day	399	99	12.1	300	15.3	0.035
2–4 times a week	561	155	19.4	406	20.7	
Once a week	464	126	15.7	338	17.2	
None	1338	421	52.6	917	46.8	
Sleep insufficiency						
Satisfied	856	202	25.5	654	33.8	<0.0001
A little dissatisfied	1279	362	45.6	917	47.4	
Very dissatisfied to quite problematic	594	229	28.9	365	18.9	
Employment change						
Yes	209	76	9.3	133	6.7	0.011
No	2597	737	90.5	1860	93.3	
Family finances						
Severe	568	207	27.3	361	19.4	<0.0001
Below average	946	278	36.6	668	36	
Average	1021	245	32.3	776	41.8	
Not severe	81	29	3.8	52	2.8	
Psychological distress						
K6 < 13	2201	588	75.1	1613	84.8	<0.0001
K6 ≥ 13	484	195	24.9	289	15.2	
Trauma symptoms						
PCL < 44	1929	494	64.5	1435	76.4	<0.0001
PCL ≥ 44	715	272	35.5	443	23.6	
Social network						
LSN_6 < 12	1183	378	48.4	805	42	0.003
LSN_6 ≥ 12	1513	403	51.6	1110	58	
Alcohol consumption (drinks)						
<4	1588	390	50.3	1198	62.8	<0.0001
≥4	1095	385	49.7	710	37.2	

**Table 5 ijerph-20-00319-t005:** Crude HRs and 95% CIs for recovery from problem drinking from FY 2012 to FY 2017.

	All		Men		Women	
	HR	95%CI	HR	95%CI	HR	95%CI
Sex (Reference: Women)						
Men	0.97	0.86–1.09				
Age (Reference: ≥65 years)						
20–49 years	0.75	0.67–0.85	0.77	0.67–0.88	0.71	0.54–0.94
50–64 years	0.78	0.70–0.86	0.78	0.70–0.87	0.77	0.57–1.05
Subjective health condition (Reference: Very good–Good)						
Normal	0.89	0.79–1.00	0.90	0.79–1.03	0.84	0.64–1.10
Poor–Very poor	0.73	0.63–0.84	0.74	0.63–0.87	0.69	0.50–0.95
History of diagnosed mental illness (Reference: No)						
Yes	0.78	0.67–0.91	0.77	0.64–0.92	0.81	0.60–1.09
Sleep insufficiency (Reference: Satisfied)						
A little dissatisfied	0.9	0.81–0.99	0.89	0.80–1.00	0.91	0.70–1.18
Very dissatisfied to quite problematic	0.72	0.63–0.82	0.69	0.59–0.80	0.81	0.61–1.07
Employment change (Reference: No)						
Yes	0.85	0.7–1.01	0.95	0.77–1.17	0.63	0.44–0.89
Family finances (Reference: Average)						
Severe	0.77	0.67–0.8	0.75	0.65–0.87	0.86	0.64–1.15
Below average	0.9	0.81–1.00	0.93	0.83–1.05	0.79	0.63–0.99
Not severe	0.89	0.67–1.17	0.91	0.67–1.26	0.8	0.43–1.47
Psychological distress (Reference: K6 < 13)						
K6 ≥ 13	0.72	0.64–0.82	0.7	0.60–0.81	0.79	0.63–1.00
Trauma symptom (Reference: PCL< 44)						
PCL ≥ 44	0.72	0.64–0.80	0.72	0.63–0.81	0.7	0.56–0.88
Social network (Reference: LSN_6 ≥ 12)						
LSN_6 < 12	0.93	0.85–1.02	0.93	0.83–1.03	0.94	0.77–1.15
Alcohol consumption (Reference: <4 drinks)						
≥4 drinks	0.77	0.70–0.84	0.77	0.70–0.86	0.72	0.57–0.92

**Table 6 ijerph-20-00319-t006:** Multivariate-adjusted HRs and 95% CIs for recovery from problem drinking from FY 2012 to FY 2017.

	All		Men		Women	
	HR	95%CI	HR	95%CI	HR	95%CI
Sex (Reference: Women)						
Men	0.98	0.87–1.11				
Age (Reference: ≥65 years)						
20–49 years	0.81	0.71–0.92	0.84	0.73–0.97	0.73	0.53–1.00
50–64 years	0.82	0.73–0.91	0.82	0.73–0.92	0.77	0.56–1.06
Subjective health condition (Reference: Very good–Good)						
Normal	0.96	0.84–1.08	0.98	0.85–1.12	0.85	0.64–1.13
Poor–Very poor	0.91	0.78–1.07	0.93	0.78–1.11	0.86	0.58–1.26
History of diagnosed mental illness (Reference: No)						
Yes	0.87	0.74–1.02	0.86	0.72–1.04	0.86	0.62–1.18
Sleep insufficiency (Reference: Satisfied)						
A little dissatisfied	0.94	0.84–1.04	0.93	0.83–1.04	1.02	0.78–1.35
Very dissatisfied to quite problematic	0.85	0.74–0.98	0.81	0.68–0.95	1.04	0.75–1.45
Family finances (Reference: Average)						
Severe	0.92	0.81–1.06	0.9	0.78–1.05	1.04	0.76–1.43
Below average	0.98	0.88–1.09	1.02	0.9–1.15	0.84	0.67–1.07
Not severe	0.87	0.65–1.15	0.87	0.63–1.2	0.84	0.45–1.58
Psychological distress (Reference: K6 < 13)						
K6 ≥ 13	0.91	0.78–1.06	0.89	0.75–1.07	0.96	0.71–1.28
Trauma (Reference: PCL < 44)						
PCL ≥ 44	0.83	0.73–0.95	0.85	0.73–0.98	0.77	0.59–1.01
Alcohol consumption (Reference: <4 drinks)						
≥4 drinks	0.77	0.70–0.85	0.77	0.69–0.86	0.73	0.57–0.93

Multivariate analysis adjusted for sex and age.

## Data Availability

The datasets analyzed during the present study are not publicly available because the data from the Fukushima Health Management Survey belongs to the government of Fukushima Prefecture, and can only be used within the organization.

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
