# Peer review of "A Six-Year Prospective Study on Problem Drinking among Evacuees of the Great East Japan Earthquake: The Fukushima Health Management Survey"

_ijerph, 2022, doi:10.3390/ijerph20010319_

Round 1

Reviewer 1 Report

Your paper very clearly lays out a thesis and defends it with empirical evidence. Because it is a longitudinal study, it explores new data. It has an important applied aspect of identifying risks for problem drinking related to trauma. 

A few areas I might elaborate on. It offers vague policy suggestions without giving any particulars to the kinds of psychological and government supports that should be given. In a few places, it would benefit from comparative data on trauma/drinking in other cultural contexts. And it would be good to place problem drinking in a wider context of substance abuse post trauma. 

I have included notes in the margins of your paper, where I think improvements can be made. 

Overall, it's a strong paper and deserves to be published. 

Author Response

Response to Reviewer 1 Comments

  • Is "problem drinking" the official term? There's a lot of cultural variability about what consitutes excessive drinking; can this be specified about Japan?

Response 1: Thank you for your important comment. A CAGE score of ≥2 is defined as problem drinking. We have explained the definition on p.5.

  • Is more than 4 drinks a day the definition of heavy drinking alone? And problem drinking is 4 drinks per day plust he other factors? This could be clarified.

Response 2: I appreciated this point, and I am very sorry for the confusion.

Yes, more than 4 drinks a day is the definition of heavy drinking. Problem drinking is defined as CAGE score of ≥2.

  • Is it just drinking or other substance abuse increases, as well? Can drinking be contexualized as an overall increase in harmful behavior after trauma?

Response 3: Thank you for this significant comment.

Numerous studies have already shown the correlation between frequent binge drinking and post-traumatic stress (North et al., 2011; Welch et al., 2014). Yet, as you have stated, only drinking cannot be contextualized as an overall increase in harmful behavior after trauma. The previous research mentioned that smoking cigarettes was higher in post-hurricane than on pre-hurricane prevalence data. Therefore, we have changed the sentence in our manuscript, in line 42.

Research has shown that post-traumatic stress after natural disasters is linked to an elevated risk of problem drinking [1] and substance use, such as cigarette use [2, 3].

  • Cite the literature on the overall rise in other harmful behavior. It's interesting that the media also framed the post-3/11 event as positive in bringing together communities, increasing sociality, and increasing marriage and birth rates.

Response 4: Thank you for this important point. We have cited the literature on other harmful behavior in our revised manuscript in lines 52- 53.

  • This Introduction is one single paragraph. I would break it into at least two paragraphs to make it easier for the reader.

Response 5:  I appreciate your valuable opinion. As you have suggested, there are now two paragraphs in the revised manuscript.

  • You may want to be exact in defining sleep insufficiency. You are very precise about problem drinking, but there is no definition or empirical data.

Response 6: I appreciated your important point out. The definition of sleep deprivation in this study was based on the data received as answers in the respondents' subjective assessment. As indicated in the results table, respondents were asked to classify their responses as "satisfied," "a little dissatisfied" and "very dissatisfied to quite problematic."

As you have mentioned, there is no definition or empirical data specifically. Therefore, we added your point of view as a limitation of our research. We have added the sentence below in lines 343-345.

Problem drinking was assessed in this study using a standard questionnaire CAGE, while sleep was assessed solely on the basis of participants' subjective symptoms.

  • Since this is the first study of its kind, you may want to highlight its innovativeness in the introduction and say WHY 6 years versus 2 years is important. You next write about implications and implying things -- but why is it at 6 years and not 7 or 5 years?

Response 7: The period was set at six years in order to ensure a long follow-up and that up-to-date data are used as much as we could at this point from the Fukushima Health Management Survey. Additionally, I have added this sentence in lines 72-75; “Most studies to date have looked at short-term impacts first 2 years following the disaster, with few looking at the longer term. We considered the importance of conducting a long-term longitudinal study for the evacuees of the Great East Japan Earthquake in 2011.”

  • What social/biological things accounts for more men than women?(16)

Response 8: The previous research has shown that severe family finances, high blood pressure, and hyperlipidemia were more likely to happen to men than womento develop problem drinking among the men evacuees from 2012 to 2013 (Ueda et al., 2019). This manuscript has concluded that men have a higher risk to develop problem drinking for a six-year follow-up. In short, men have a higher risk to develop problem drinking because of social and biological factors.

  • Long paragraph -- consider breaking it into two paragraphs.

Response 9: I appreciate your valuable opinion. As you have suggested, there are now two paragraphs in the revised manuscript.

  • Conclusion is a bit short.

Response 10: Thank you very much for your important comment to develop our conclusion. We added some sentences in red in the manuscript.

Reviewer 2 Report

The paper is an excellent study, dealing with issues that could be applied to the disaster evacuees of any geographical region in the world. This is a profound research with all necessary requirements for a serious publication - meaningful introduction, strong explanation of used materials and methods, proven results, a lot of data tables illustrating the research, valuable appendixes, a logic flow of the presentation.

This is a must read for all professionals in the field.

Author Response

I am very appreciated for your kind comments. I have responded all your queries